# Dreaming with hippocampal damage

**Goffredina Spanò[1], Gloria Pizzamiglio[1], Cornelia McCormick[2], Ian A Clark[1], Sara De Felice[1], Thomas D Miller[3], Jamie O Edgin[4], Clive R Rosenthal[5], Eleanor A Maguire[1]\***

[1]Wellcome Centre for Human Neuroimaging, UCL Queen Square Institute of Neurology, University College London, London, United Kingdom; [2]Department of Neurodegenerative Diseases and Geriatric Psychiatry, University Hospital Bonn, Bonn, Germany; [3]Department of Neurology, Royal Free Hospital, London, United Kingdom; [4]Department of Psychology, University of Arizona, Tucson, United States; [5]Nuffield Department of Clinical Neurosciences, University of Oxford, Oxford, United Kingdom

**Abstract** The hippocampus is linked with both sleep and memory, but there is debate about whether a salient aspect of sleep – dreaming – requires its input. To address this question, we investigated if human patients with focal bilateral hippocampal damage and amnesia engaged in dreaming. We employed a provoked awakening protocol where participants were woken up at various points throughout the night, including during non-rapid eye movement and rapid eye movement sleep, to report their thoughts in that moment. Despite being roused a similar number of times, dream frequency was reduced in the patients compared to control participants, and the few dreams they reported were less episodic-like in nature and lacked content. These results suggest that hippocampal integrity may be necessary for typical dreaming to occur, and aligns dreaming with other hippocampal-dependent processes such as episodic memory that are central to supporting our mental life.

## Introduction

Dreaming has intrigued humans for thousands of years, being variously interpreted as having premonitory, religious, psychoanalytic, or mnemonic significance. Defined as an internally-generated subjective mental experience during the sleep state (*Cipolli et al., 2017*), dreaming can be present at initial sleep onset (hypnagogic sleep; *Horikawa et al., 2013*; *Stickgold et al., 2000*), during non-rapid eye movement (NREM) sleep (*Antrobus et al., 1995*; *Foulkes, 1962*; *Nielsen, 2000*; *Siclari et al., 2017*; *Wamsley, 2013*; *Wamsley et al., 2007*), and rapid eye movement (REM) sleep (*Hobson et al., 2000*; *Maquet et al., 2000*; *Maquet et al., 2005*).

Although dreams are not a precise replay of our memories (*Fosse et al., 2003*; *Stickgold et al., 2001a*), it has been proposed that dreaming may be associated with memory consolidation processes (*Payne, 2010*; *Wamsley, 2014*; *Wamsley et al., 2010*; *Wamsley and Stickgold, 2019*). Indeed, in rodents and humans, patterns of brain activity exhibited during a learning experience were found to be subsequently expressed during sleep (*Peigneux et al., 2004*; *Wilson and McNaughton, 1994*).

Bilateral damage to a brain structure called the hippocampus is known to adversely affect memory processing (*Miller et al., 2017*; *Miller et al., 2020*; *Scoville and Milner, 1957*; *Spiers et al., 2001*) and daydreaming (*McCormick et al., 2018*). While sleep dreaming has been examined previously in patients with hippocampal lesions, results are mixed, with some studies reporting repetitious and stereotyped dreams (*Torda, 1969a*; *Torda, 1969b*; *Stickgold et al., 2000*), whereas others claim that hippocampal damage has no effect on dreaming (*Solms, 1997*; *Solms, 2013*). Possible reasons for these disparate findings include differences in the sleep stages sampled, the inclusion in

**\*For correspondence:**
e.maguire@ucl.ac.uk

**Competing interests:** The authors declare that no competing interests exist.

**eLife digest** Dreaming has intrigued humans for thousands of years, but why we dream still remains somewhat of a mystery. Although dreams are not a precise replay of our memories, one idea is that dreaming helps people process past experiences as they sleep. If this is true, then part of the brain called the hippocampus that is important for memory should also be necessary for dreaming.

Damage to the hippocampus can cause a condition called amnesia that prevents people from forming new memories and remembering past experiences. However, studies examining dreaming in people with amnesia have produced mixed results: some found that damage to the hippocampus had no effect on dreams, while others found it caused people to have repetitive dreams that lacked detail. One reason for these inconsistencies is that some studies asked participants about their dreams the next morning by which time most people, particularly those with amnesia, have forgotten if they dreamed.

To overcome this limitation, Spanò et al. asked participants about their dreams immediately after being woken up at various points during the night. The experiment was carried out with four people who had damage to both the left and right hippocampus and ten healthy volunteers. Spanò et al. found that the people with hippocampal damage reported fewer dreams and the dreams they had were much less detailed.

These findings suggest that a healthy hippocampus is necessary for both memory and dreaming, reinforcing the link between the two. Hippocampal damage is associated with a number of diseases, including dementia. If these diseases cause patients to dream less, this may worsen the memory difficulties associated with these conditions.

---

several studies of patients with psychiatric conditions and below-average IQ and, in other instances, patients were interviewed at a point temporally remote from any dreaming that might have occurred, presenting a challenge for these patients who usually suffer from amnesia.

It remains uncertain, therefore, whether hippocampal integrity is necessary for dreaming to occur. If, as we predicted, bilateral hippocampal damage degrades dreaming, this would reinforce the link between dreaming and hippocampal-dependent processes such as memory, potentially moving us closer to an understanding of why we dream. By contrast, if hippocampal-damaged patients who have a reduced ability to experience complex, imagery-rich, spatio-temporal mental events during wake are nevertheless able to have such experiences during sleep, this would need to be explicitly accounted for within theories of hippocampal function.

Here, we sought to mitigate the issues affecting previous studies while investigating the frequency, features and content of dreaming in rare patients with selective bilateral hippocampal damage and matched control participants. To do this we employed a provoked awakening protocol (*Nguyen et al., 2013*; *Siclari et al., 2013*; *Wamsley et al., 2016*). This widely-used approach involved waking participants up at various times during their night's sleep to report their thoughts in that moment. In this way we examined sleep mentation in a direct and immediate manner.

## Results

We assessed four patients (all right-handed males; mean age 58.25 years, SD ±20.82) with selective bilateral hippocampal lesions and a specific episodic memory deficit (*Table 1*, Material and methods, *Supplementary file 1*, *Spanò et al., 2020*). Patients were matched to ten healthy control participants (all right-handed males; mean age 59.2 years, SD ±15.89) based on a number of demographic factors including age, gender, body mass index, non-verbal IQ, and a range of sleep quality measures (*Table 1*; Material and methods, *Supplementary file 1*, *Supplementary file 2*). We conducted in-home sleep recording using portable polysomnography (PSG) on two consecutive nights – one habituation night to allow for familiarization with the PSG equipment and one experimental night for the collection of dream reports. The purpose of the PSG recording was to ensure that we awakened participants during both NREM and REM sleep, and in a similar manner for the patient and control groups. During the dream sampling night, participants were woken up after a period of 3 min from

**Table 1.** Demographic characteristics.

| Group | Age (years) | Chronicity (years) | LHPC volume (mm³) | RHPC volume (mm³) | LHPC % volume loss relative to CTL[a] | RHPC % volume loss relative to CTL[a] | WASI |
|---|---|---|---|---|---|---|---|
| CTL | 59.20 (15.89) | n.a. | 3173.18[a] (338.89) | 3285.91[a] (300.81) | n.a. | n.a. | 14.50 (2.37) |
| HPC1 | 61 | 6 | 2506 | 2803 | −21.03% | −14.70% | 12 |
| HPC2 | 72 | 8 | 1736 | 1698 | −45.29% | −48.32% | 10 |
| HPC3 | 72 | 11 | 2607 | 2755 | −17.84% | −16.16% | 12 |
| HPC4 | 28 | 11 | 2819 | 2804 | −11.16% | −14.67% | 14 |

All patients (HPC1-4) and control participants (CTL) were right-handed males. Mean and standard deviation in parentheses are shown for control participants and individual data for the four patients. [a]The control group consisted of eleven participants (mean age 55.64 years ± 16.47). LHPC = left hippocampus; RHPC = right hippocampus; n.a. = not applicable; WASI = Wechsler Abbreviated Scale of Intelligence (**Wechsler, 1999**) Matrix Reasoning subtest scaled score. See **Supplementary file 1** and **Supplementary file 2** for additional neuropsychological and sleep quality data of the participants.

the onset of either NREM or REM sleep at various times throughout the night (Materials and methods, *Figure 1A*).

*Table 2* shows the group summary data and the results of the between-group statistical analyses for each of the measures that are described below. While, for the sake of economy, we present the findings in terms of these group comparisons, given the small sample of these rare patients, caution should be exercised in interpreting the results. We, therefore, include the individual patient data in *Tables 1* and *2* permitting the patients to be considered also as a series of case studies.

The number of awakenings was not different between patients and control participants (*Figure 1B*). Furthermore, there were no significant group differences in the proportion of awakenings from NREM and REM sleep (*Figure 1C*). After an awakening, participants were instructed via a two-way intercom to describe everything that was going through their mind before they were woken up in that moment. They were occasionally probed (e.g. Can you tell me more about that?) to obtain further information (Materials and methods). The amount of probing did not differ between groups. Dream reports were subsequently transcribed for further analyses (Materials and methods; *Figure 2*).

Although the two groups were woken up a similar number of times, they differed in terms of dream frequency, with patients reporting significantly fewer dreams compared to control participants (*Figure 1D*). There were no group differences in terms of the proportion of dream recall during NREM and REM sleep. Of note, one patient did not report any dreams at all, but instead typically stated 'I can't picture it' (*Figure 2*).

Perhaps the patients reported fewer dreams than control participants simply because they forgot any dreams they may have had. Three different types of awakening were evident: when participants reported a dream, when they stated they did not dream at all (no dream), and when they dreamt but could not recall the content (this is known as a blank dream). Although there was a significant difference between patients and controls for dreams (see above) and for no dreams, they did not differ in terms of the proportion of blank dreams, which was low for both groups. This suggests that patients could distinguish between situations when they did not dream and those when they dreamt but could not remember. However, no validated objective measure of dreaming exists, and this should be borne in mind when interpreting participants' subjective reports. It is notable that in previous studies, these particular patients could retain information over several minutes, including reporting on their daydreaming (e.g. *McCormick et al., 2016*; *McCormick et al., 2018*), which speaks against a rapid decay of sleep mentation as an explanation for their reduced dream frequency.

The patients had so few dream reports that comparisons with the control participants should be treated with caution. Nevertheless, we wondered whether any differences in features and content were evident between the groups for the few dreams the patients reported. Because one patient had no dreams whatsoever, he was not included in these analyses.

We first examined the number of informative words (see Materials and methods; *Stickgold et al., 2001b*) used in the dream narratives and, while the patients used fewer such words, overall there was no significant difference between the groups. Nevertheless, adjudicating between the possibility of patients having a generic problem with expressing themselves verbally versus merely having little

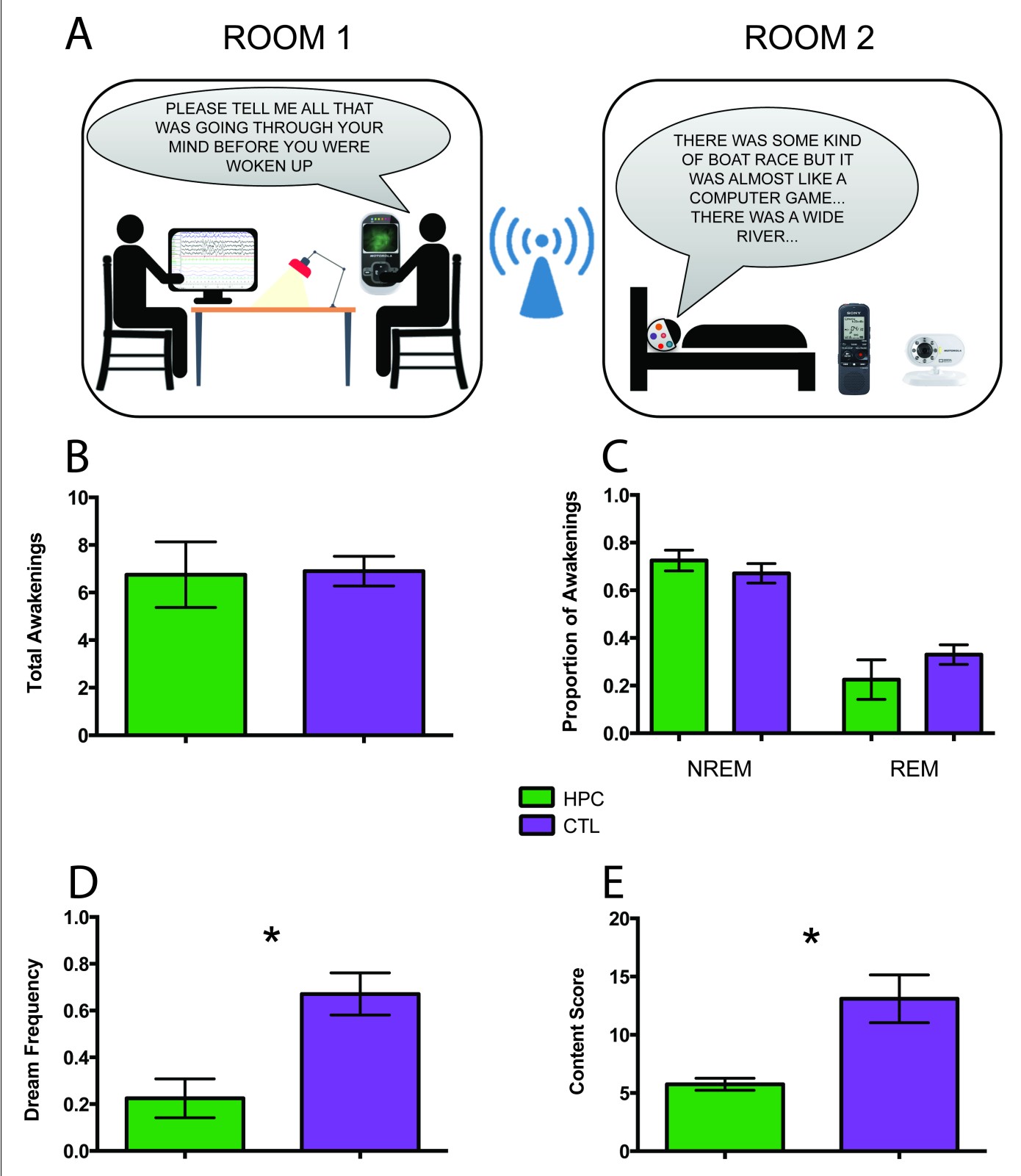

**Figure 1.** Experimental set-up and key findings. (**A**) Two researchers were located in Room one which was adjacent to Room two where the participant slept. The participant was woken up at various times during their night's sleep to report their thoughts in that moment. PSG recordings informed the decisions about when to awaken the participant to ensure sampling during non-rapid eye movement (NREM) and rapid eye movement (REM) sleep. We used a Bluetooth intercom system equipped with a camera for continuous visual monitoring and communication with the participant. (**B**) The number of

*Figure 1 continued*

total awakenings was not different between the patients (HPC) and control (CTL) participants. (C) There were also no significant group differences in the proportion of awakenings from NREM and REM sleep. (D) In contrast, the patients reported significantly fewer dreams than the control participants, expressed here as the total number of dreams divided by the total number of awakenings (+ / - 1 SEM; p=0.028). (E) The few dreams the patients had were significantly less rich in content compared to those of the control participants (n = 3 patients, as one patient had no dreams at all and was not included in this analysis; + / - 1 SEM; p=0.018). For other measures see *Table 2*.

to describe because the few dreams they had were so impoverished, is challenging. The same issue pertains for assessments during wake. A number of studies addressed this concern during tasks involving the imagination of scenes or future scenarios, counterfactual thinking, describing the present and pictures of scenes. Bilateral hippocampal damage does not affect narrative construction or verbal descriptive ability (e.g. *Race et al., 2011*; *Race et al., 2013*; *Mullally et al., 2012*; *Mullally and Maguire, 2014*; *Miller et al., 2020*). Considering specifically the patients in the current

**Table 2.** Dream characteristics.

| | HPC M (SD) | CTL M (SD) | U | ES | P-Value | HPC1 | HPC2 | HPC3 | HPC4 |
|---|---|---|---|---|---|---|---|---|---|
| *General analyses[a]* | | | | | | | | | |
| Number of awakenings | 6.75 (2.75) | 6.90 (1.97) | 18.5 | 0.11 | 0.829 | 10.00 | 4.00 | 5.00 | 8.00 |
| Proportion of awakenings during NREM | 0.73 (0.09) | 0.67 (0.13) | 14.0 | 0.47 | 0.383 | 0.80 | 0.75 | 0.60 | 0.75 |
| Proportion of awakenings during REM | 0.23 (0.17) | 0.33 (0.13) | 13.0 | 0.55 | 0.309 | 0.00[c] | 0.25 | 0.40 | 0.25 |
| Number of probes per awakening | 3.82 (1.75) | 4.22 (1.13) | 17.0 | 0.23 | 0.671 | 5.80 | 3.50 | 1.60 | 4.38 |
| Dream frequency | 0.23 (0.17) | 0.67 (0.28) | 4.5 | 1.45 | **0.028** | 0.40 | 0.25 | 0.00 | 0.25 |
| Proportion of dreams during NREM | 0.38 (0.48) | 0.52 (0.20) | 14.5 | 0.43 | 0.415 | 1.00 | 0.00 | 0.00 | 0.50 |
| Proportion of dreams during REM | 0.38 (0.48) | 0.48 (0.20) | 16.0 | 0.31 | 0.555 | 0.00 | 1.00 | 0.00 | 0.50 |
| Proportion of no dreams | 0.65 (0.31) | 0.21 (0.24) | 4.0 | 1.52 | **0.022** | 0.60 | 0.75 | 1.00 | 0.25 |
| Proportion of blank dreams | 0.13 (0.25) | 0.12 (0.15) | 15.5 | 0.35 | 0.496 | 0.00 | 0.00 | 0.00 | 0.50 |
| *Overall qualitative attributes[b]* | | | | | | | | | |
| Number of informative words | 43.17 (16.06) | 95.55 (55.20) | 7.0 | 0.81 | 0.176 | 55.50 | 25.00 | . | 49.00 |
| Complexity | 2.67 (0.58) | 3.32 (0.70) | 5.5 | 1.00 | 0.098 | 3.00 | 2.00 | . | 3.00 |
| Vividness | 3.10 (0.79) | 3.88 (1.08) | 7.5 | 0.75 | 0.203 | 2.80 | 4.00 | . | 2.50 |
| Bizarreness | 1.58 (1.01) | 2.27 (0.95) | 8.0 | 0.70 | 0.232 | 2.75 | 1.00 | . | 1.00 |
| Emotional valence | 2.75 (0.25) | 2.81 (0.22) | 13.0 | 0.19 | 0.720 | 2.75 | 3.00 | . | 2.50 |
| Proportion of self-references | 0.84 (0.29) | 0.90 (0.19) | 15.0 | 0.00 | 1.000 | 1.00 | 1.00 | . | 0.50 |
| *Content characterization[b]* | | | | | | | | | |
| Internal (episodic) details | 4.08 (1.47) | 9.13 (3.56) | 2.0 | 1.54 | **0.028** | 5.75 | 3.00 | . | 3.50 |
| External (semantic/other) details | 0.17 (0.29) | 1.07 (1.60) | 8.5 | 0.64 | 0.258 | 0.00 | 0.00 | . | 0.50 |
| Content score | 5.75 (0.90) | 13.10 (6.49) | 1.0 | 1.74 | **0.018** | 6.75 | 5.00 | . | 5.50 |

M = mean; SD = standard deviation; ES = effect size; HPC = hippocampal-damaged patients; CTL = control participants; NREM = non-rapid eye movement sleep; REM = rapid eye movement sleep; HPC1–4 = each individual hippocampal-damaged patient. P-values relate to between-group non-parametric Mann-Whitney U tests with significant differences depicted in bold. [a]All patients included; means are per awakening. [b]HCP3, who had no dream reports at all, was excluded; means are per dream report. [c]For HCP1, during 20% of his awakenings towards the end of the night, the EEG cap stopped functioning and so designation to NREM or REM sleep was not possible. Hence, it could be that this zero score for REM awakenings is an underestimate, given that REM is more common in the latter part of the night. Note that his dream reports from these awakenings were still included in the dream qualitative attributes and content analyses. See *Table 2—source data 1* file for the data underpinning this table.

The online version of this article includes the following source data for Table 2:

Source data 1. This file contains the individual participant data for every dream-related measure that is summarised in *Table 2*.

**HPC1:** I was just walking up the railway line. *Can you describe it to me? How did it look to you?* Just like an old…um…it was just like a flat area. And the track was just there. And I was walking up…there was nothing very descriptive than just being fairly flat.

**HPC2:** I was dreaming I was in a nightclub. *Can you tell me a little bit more about it?* A nightclub. I was drinking. *What was happening around you? What were you doing?* People were dancing. *Anything else?* No that's it.

**HPC3:** Erm…I can't picture it. It was…I think I was trying to get comfy…I was partway through a turn when you…yeah that was it. Trying to get this belt pushed out of my way so that I could turn over. *So was it just thoughts or were you visualising anything?* No, no. Nothing. Just…that was just, I was partway through a turn. I was just trying to move a little bit. That was all. Thank you.

**HPC4:** I think I was dreaming about stuff related to work…um…I think something that was related to last week, but I…a lot of it was like conversations that haven't happened. *What was happening around you, what were you doing?* It was a conversation happening in the locker room. *Can you give us any more specifics about the conversations or anything else that you recall?* They've decided they're changing the location of the locker room so…um…people were not very happy about it…um…so they were having conversations around why things were happening a certain way and how it can be done differently. *People that were there were colleagues that are your work colleagues?* No, so there were people, but probably not, if I'm honest with you. It was just random people.

**CTL:** There was an action for the Sunday Times…um…and I just said, "Oh good. Somebody can do something with it and make it better". So they were selling it to another company and separating it from the main Times daily newspaper. And the bidding was just starting…and it was in a…I'm not quite sure where it was but it was in a big…it's a bit like a restaurant type pub type building…and it was packed…and they were literally just about to start the bidding for the paper. *Anything else, any other detail?* I remember there were screens that everybody was looking at but I'm not quite sure what they had on them…um…presumably it was going to be the amount of the bid each time…but there was more detail on there thinking about it…maybe it was different newspapers…and the Sunday Times was one of them…yes, the Sunday Times was written on one of the screens…um…but I can't remember much before that…but literally the auction was just about to begin…for the Sunday Times…a bit bizarre!

**Figure 2.** Example dream reports. Experimenter probing is shown in italics. HPC1−4 = the four hippocampal-damaged patients; CTL = an example control participant.

study, they too had no difficulty performing verbal fluency tests or other tasks where total word count was measured (*Supplementary file 1*). Given these general and specific findings, it is unlikely that the patients' performance was driven by an underlying expressive verbal problem.

Next we assessed overall qualitative features of dream reports using experimenter ratings (Materials and methods; *De Gennaro et al., 2011*; *Oudiette et al., 2012*). There were no significant group differences in terms of general complexity, vividness, bizarreness, emotional valence (there were no nightmares), and proportion of self-references. In order to probe the dream reports in more depth, we used two scoring methods that are often employed for examining complex mental events (Materials and methods). The first involved a scoring regime typically used for autobiographical memories, the Autobiographical Interview (*Levine et al., 2002*). This allowed us to measure the amount of episodic (internal) and non-episodic (external) details. The patients included significantly fewer internal details in their dream narratives relative to controls, whereas there was no significant group difference in terms of external details. A second method, usually employed for scoring the

content of imagined scenes (*Hassabis et al., 2007*), showed that the dream reports of patients were significantly less rich in content compared to those of the control participants (*Figure 1E*).

## Discussion

By studying rare patients with selective bilateral hippocampal damage we found that dream frequency was reduced compared to control participants, and the few dreams they had were less episodic-like in nature and lacked content. This accords with previous studies that reported stereotyped dreaming in patients with brain damage that extended beyond the hippocampi (*Torda, 1969a*; *Torda, 1969b*; *Stickgold et al., 2000*), and echoes the effects of hippocampal damage on imagination (*Hassabis et al., 2007*), episodic memory (*Miller et al., 2017*; *Miller et al., 2020*; *Spiers et al., 2001*) and daydreaming (*McCormick et al., 2018*). Given our patients' circumscribed lesions, these results suggest that hippocampal integrity may be necessary for typical dreaming to occur. There are several possible explanations for degraded dreaming in the patients, which we consider in turn.

Despite the tendency to confine dreaming to the sleep state, studies have shown that there is often continuity between waking life experiences and dreaming (*Andrillon et al., 2015*; *Fosse et al., 2003*; *Horikawa et al., 2013*; *Schredl and Hofmann, 2003*). Given that the patients had difficulty retaining information over longer time scales while awake, then perhaps during subsequent sleep there was little material to process. Hence it could be that the capacity to dream was intact, but underused, in the patients.

Alternatively, the core capacity for dreaming might have been compromised, with this in turn affecting memory processing. Dreaming has been linked with the consolidation of information into long-term memory (*Payne, 2010*; *Wamsley, 2013*; *Wamsley, 2014*; *Wamsley et al., 2010*; *Wamsley and Stickgold, 2019*), but it is unknown whether dreaming plays a functional role in this process (*Wamsley, 2014*). If it does, then patients with hippocampal damage may lack this dream-related mechanism for facilitating memory processing, and this could contribute to their amnesia.

Another aspect of sleep that has been associated with episodic memory consolidation is slow-wave sleep (SWS), a stage within NREM sleep (*Rasch and Born, 2013*). We have previously shown that the hippocampal-damaged patients tested here had significantly reduced SWS and slow wave activity, whereas the time spent in other sleep stages was comparable to that of control participants (*Spanò et al., 2020*). It is unlikely that the patients' degraded dreaming was caused solely by decreased SWS. This is because we sampled dreaming across both NREM and REM sleep, and of the dreams sampled during NREM sleep in the control participants, just 5% (SD 11.3) were during SWS. Overall, however, reduced slow wave activity along with sub-optimal dreaming – if dreaming plays a functional role – may constitute a twofold blow that adversely affected the proper functioning of episodic memory in these patients.

A different perspective on the current results involves looking beyond memory. The hippocampus has been implicated in a range of other cognitive functions including thinking about the future (*Addis et al., 2007*; *Hassabis et al., 2007*; *Kurczek et al., 2015*), spatial navigation (*Maguire et al., 2006*; *O'Keefe and Nadel, 1978*), daydreaming (*Karapanagiotidis et al., 2017*; *McCormick et al., 2018*), aspects of decision making (*Mullally and Maguire, 2014*; *McCormick et al., 2016*), and visual perception (*Lee et al., 2005*; *McCormick et al., 2017*; *Mullally et al., 2012*). Patients with bilateral hippocampal damage are also impaired at mentally constructing scene imagery (*Hassabis et al., 2007*), and scene imagery features prominently across cognition (*Clark et al., 2019*; *Clark et al., 2020*). Consequently, it has been proposed that one role of the hippocampus could be to facilitate the scene construction process (*Hassabis and Maguire, 2007*; *Maguire and Mullally, 2013*). This may explain why dreaming is degraded in the context of hippocampal pathology, as it too typically involves (and possibly requires) scene imagery.

In summary, while the functional role of dreaming is as yet unknown, we conclude from our results that hippocampal integrity may be a prerequisite for typical dreaming to occur, and that dreaming seems to align with an array of other cognitive functions that are hippocampal-dependent and which play crucial roles in supporting our everyday mental life.

# Materials and methods

## Participants

For all patients, hippocampal lesions resulted from leucine-rich glycine-inactivate-1 antibody-complex limbic encephalitis (LGI1-antibody-complex LE; *Miller et al., 2017*; *Miller et al., 2020*). This study was conducted a median of 9.5 years after hippocampal damage occurred (mean 9 years ± SD 2.45). Patients (HPC) and the dream control participants (CTL) were closely matched on a number of demographic factors: gender (all males), age (*Table 1*; MWU = 19.00, p=0.89, Cohen's $d$ = 0.08), body mass index (HPC mean 27.68 ± 2.51; CTL 25.79 ± 2.41; MWU = 14.00, p=0.40, Cohen's $d$ = 0.47) and general cognitive ability assessed with the Matrix Reasoning subtest of the Wechsler Abbreviated Scale of Intelligence (WASI; *Wechsler, 1999*; *Table 1*; MWU = 7.00, p=0.06, Cohen's $d$ = 1.13). Participants had no history of psychiatric disorders (e.g., depression, anxiety). Each participant gave written informed consent for participation in the study, for data analysis and for publication of the study results. 'Materials and methods' were approved by the University College London Research Ethics Committee.

The patients entered the sleep study having already been characterized, relative to matched healthy control participants, in terms of their lesion selectivity and neuropsychological profile as part of previous research studies. Full details of that characterization process are available in *McCormick et al., 2016*, *McCormick et al., 2017*, *McCormick et al., 2018* and *Spanò et al., 2020*. In summary, manual (blinded) segmentation of the hippocampi from T2-weighted high resolution structural MRI scans (0.5 × 0.5 × 0.5 mm voxels) showed that our patients had substantial volume loss relative to controls in the left (MWU = 2.00, p=0.009, Cohen's $d$ = 1.83) and right (MWU = 3.00, p=0.013, Cohen's $d$ = 1.67) hippocampus (*Table 1*). Expert neuroradiological examination confirmed there was no damage outside of the hippocampi. In addition, automated whole brain voxel-based morphometry showed there were no volume differences between patients and controls anywhere else in the brain. *Supplementary file 1* provides the neuropsychological profile (summary data and statistical analyses) of the patients across a range of cognitive tests, and indicates the selective nature of their memory loss.

Since all patients included in the current study had suffered from LGI1-antibody-complex LE, our findings might potentially not generalize to other forms of hippocampal amnesia. However, it is important to note that other aetiologies that lead to hippocampal-mediated amnesia such as viral encephalitis, hypoxic brain injury secondary to drug overdose, or toxic shock syndrome are associated with circumscribed hippocampal lesions, but frequently also involve anatomical damage elsewhere (*Heinz and Rollnik, 2015*; *Raschilas et al., 2002*). In addition, these aetiologies lead to co-morbidities and broader cognitive impairment (*Heinz and Rollnik, 2015*; *Hokkanen and Launes, 2007*; *McGrath et al., 1997*; *Peskine et al., 2010*; *Thakur et al., 2013*; *Rosene et al., 1982*), which were absent from the clinical and neuropsychological profile of the patients reported here. Therefore, the selection of such a rare group of patients with circumscribed hippocampal lesions allowed us to pinpoint the direct role of the hippocampus in dreaming without the interference of potential confounds associated with heterogeneity in aetiology.

Other features associated with LGI1-antibody-complex LE in its initial presentation – such as focal seizures and hyponatraemia related to hypothalamic damage – are also unlikely to explain the effects we observed. Our patients were seizure-free when they were discharged after initial admission, they were not prescribed antiepileptic medication, and none of the patients had seizure recurrence following initial treatment. Thus, unlike in temporal lobe epilepsy, which is associated with ongoing seizures and hippocampal sclerosis (*Kapur and Prevett, 2003*), our patients enabled us to study effects on dreaming that were not coincidental with, and sequelae of, seizure activity. Moreover, patients were not undergoing treatment for hyponatraemia, which is consistent with published evidence that persistent hyponatraemia is not a characteristic feature of LGI1-antibody-complex LE (*Bastiaansen et al., 2017*). Therefore, the findings in the current study are unlikely to stem from the above-mentioned potential issues.

As dreaming might be influenced by sleep quality (*Schredl, 2009*), we confirmed that patients did not differ significantly from control participants on subjective measures of the general quality and pattern of sleep, as well as on objective measures of sleep-related breathing disorders, and sleep-wake patterns across one week (see *Supplementary file 2*). Specifically, participants completed standardized questionnaires assessing habitual sleep habits over the last month (The

Pittsburgh Sleep Quality Index; *Buysse et al., 1989*), level of daytime sleepiness (The Epworth Sleepiness Scale; *Johns, 1991*), and chronotype – whether someone is a 'morning' or an 'evening' type of person (The Morningness-Eveningness Questionnaire; *Horne and Ostberg, 1976*). Moreover, we assessed the severity of obstructive sleep apnoea with the WatchPAT-200 (Itamar Medical Ltd., Caesarea, Israel), a wrist-worn device that measures the Peripheral Arterial Tone (PAT) signal by means of a plethysmographic based finger-mounted probe. Signals were automatically analysed with the zzzPAT software (version 4.4.64 .p, Itamar Medical Ltd., Caesarea, Israel) to identify respiratory events and sleep states. The outcome measure employed in this study was the PAT apnoea-hypopnea index (AHI), which provides the number of apnoea and hypopnea events per hour during the night.

In order to assess sleep-wake patterns, participants wore an Actiwatch 2 (Phillips Respironics Mini-Mitter) for seven consecutive days and nights on their non-dominant wrist. Light and activity data were collected in 30 s epochs and analyzed using the Philips Actiware 6.0.2 software package (Respironics Actiware 6.0.2). Data were scored based on available guidelines (*Chow et al., 2016*), with a medium sensitivity (40 activity cpm), with sleep onset occurring after an immobility period of 10 min, and rise time following an increase in activity level and in light level above 1.0 µW/cm$^2$. Variables of interest were sleep efficiency (in percent), total sleep time (in minutes), sleep fragmentation index (an index of restlessness), night-to-night variability for sleep duration (*Lemola et al., 2013*), average bedtime and mean sleep midpoint (clock time halfway between bedtime and rise time).

## Procedure

Participants slept in the same room on both nights. Two sleep researchers were located in a separate, nearby, room one of whom performed sleep staging in real time during online visualization of noise-reduced EEG recordings. An independent, registered sleep technologist, blind to participant group membership and the study aims, later off-line scored the PSG recordings to verify the sleep staging, in line with the revised American Academy of Sleep Medicine manual (*Berry et al., 2015*).

For all participants, awakenings occurred after sleep onset, throughout the night. We aimed to assess dreams from both NREM and REM sleep, and we therefore staggered awakenings at intervals that allowed for entry into these stages, or approximately between 30–90 min intervals. For example, a participant may have been woken at 30 min, returned to sleep and was woken again at 90 min. A maximum of 10 awakenings were scheduled per night. Awakenings were not collected at pre-ordained points during the night (for example, after the third REM period for all participants), rather they were based on participants' specific sleep architecture in order to maximize the number of reports collected. Every participant had a different sleep onset and duration, and so the awakenings were not scheduled at precisely the same clock time across participants. However, as shown in *Supplementary file 2*, the sleep quality of the patients and the controls was well-matched, and this included total sleep time, bedtime and midpoint of the night.

Once a decision to awaken a participant was made, after a 3 min period without stage shift of either NREM or REM sleep, the other sleep researcher played a non-stressful 500 Hz neutral tone via a two-way, Bluetooth intercom system equipped with a camera for continuous visual monitoring. After the tone was played, if the participant did not wake up, his name was spoken. This two-step awakening procedure was repeated up to five times, if required (*Dumel et al., 2015*). After awakening, participants were instructed by intercom to tell the experimenter everything that was going through their mind before they were woken up. What they said was recorded and subsequently transcribed.

Participants were occasionally probed (e.g. Can you tell me more about that?) to obtain further information. Probing followed a structured protocol. This involved first asking participants to freely describe what was in their minds immediately after awakening. Whenever a participant's response was not clear or only covered parts of the dream, the experimenter asked general follow-up questions, which could echo information already provided (e.g. '...It was a conversation happening in the locker room'. *Can you give us any more specifics about the conversations or anything else that you recall?*). Crucially, this probing never involved leading the participant, as can be observed in the examples provided in *Figure 2*. This approach is very similar to that of well-established tasks that assess autobiographical memory recall (*Levine et al., 2002*) and scene imagination ability (*Hassabis et al., 2007*) during wake, where probing in this manner is widely accepted.

## Dream analyses

Transcriptions of dream reports were analysed by a researcher who was blind to participant group membership. Double scoring was performed on 20% of the data by a second researcher. We assessed across-experimenter agreement with inter-class correlation coefficients, with a two-way random effects model looking for absolute agreement, which indicated excellent agreement between the experimenters' scoring (range: 0.9–1.0).

A dream was defined as any report that included at least one person, one place or one event (*Foulkes and Rechtschaffen, 1964*). Dream frequency was calculated as the total number of dreams divided by the total number of awakenings. Word count included words that provided information about the dream (i.e., informative words), and excluded repetitions, hesitations and fillers, secondary elaborations and metacognitive statements (*Stickgold et al., 2001b*). Dream complexity was experimenter scored on a 6-point scale using an adaptation of the Orlinsky score (*Oudiette et al., 2012*; excluding the no dream and blank dream options, which we recorded separately). This ranged from a participant remembering a specific topic, but in isolation, for example a fragmentary action, scene, object, word, or idea unrelated to anything else, to a participant remembering an extremely long and detailed dream sequence of five or more stages. Vividness referred to the clarity and detail of a dream, and was experimenter scored using the 6-point scale of *De Gennaro et al., 2011*, ranging from no image at all, to perfectly clear and as vivid as normal vision. Bizarreness/implausibility of dreams was experimenter scored using the *De Gennaro et al., 2011* 6-point scale. This was based on the presence of bizarre elements (impossible characters or actions) and/or improbable plot (discontinuity or unusual settings). The emotional valence of a dream was scored using a 5-point Likert scale ranging from very negative to very positive, with three indicating a neutral tone. For self-references, one point was awarded per dream report if a participant reported he was the agent of an action, thought or feeling (e.g. 'I was driving my car down a nearby road with a friend...").

We used two other scoring methods that are often employed for examining complex mental events in order to probe the dream reports further. The first was the Autobiographical Interview which identifies internal and external details, a distinction that can be conceptualized as the difference between episodic and non-episodic/semantic information, respectively (*Levine et al., 2002*). Internal details refer to the main event described by the participant and comprises the subcomponents: Event (happenings, individuals present, weather conditions, physical/emotional actions, reactions in others); Time (time of the day, year, season, month, day of the week, hours); Place (location of the event); Perceptual (auditory, olfactory, tactile, visual and visual details, body position, duration of time); Thought/emotion (emotional states, thoughts, implications). The internal details score is the sum of these subcomponents. Anything tangential to the main dream event was scored as external details, including Event (details from other events outside of the dream); Semantic (general, ongoing, extended knowledge/event/state of being); Repetition; Other (metacognitive statements, editorializing). The external details score is the sum of these subcomponents.

A second method we employed, the Scene Construction Test (*Hassabis et al., 2007*), is usually used for scoring the content of imagined scenes. Here we focused on the content score, which comprises four subcomponents: Entities Present (objects or people); Spatial References (places or spatial relationships between entities); Sensory Descriptions (details that describe an entity); Thought/emotion/action (thoughts, emotional states, action descriptions). The content score is the sum of these subcomponents.

## Sleep physiology

All participants underwent PSG in their homes using a Brain Products system (GmbH, Gilching, Germany). The purpose of the PSG recording was to ensure that we awakened participants during both NREM and REM sleep, and in a similar manner for the patient and control groups. Two trained sleep researchers arrived at a participant's home approximately three hours before the usual bedtime to set up for the PSG. Equipment was then removed by a researcher the following morning upon awakening. PSG was recorded using a 24-electrode cap (EasyCap; based on the international 10–20 system) including the following EEG channels: Fp1, Fp2, F3, F4, C3, C4, P3, P4, O1, O2, F7, F8, T7, T8, P7, P8, Fz, Cz, Pz, Oz, FT9, FT10 referenced to average mastoids (M1 and M2) (sampling rate = 500 Hz). This montage also included two bipolar electrooculogram channels (EOG), two electromyogram

channels (EMG) and two electrocardiogram channels (ECG). Sleep staging was performed based on EOG, EMG and the following derivations: F3/M2, F4/M1, C3/M2, C4/M1, O1/M2, O2/M1.

## Statistical analyses

All statistical analyses were performed with SPSS 25.0 (IBM Corporation). Given that the data did not meet the assumptions of normality and homogeneity necessary for parametric statistics, between-group analyses were performed using non-parametric Mann-Whitney U tests. We also calculated the effect sizes using non-parametric Cohen's $d$ for all tests performed. In all analyses, the significance level was set at 0.05.

# Acknowledgements

We thank the participants and their families for welcoming us into their homes and giving of their time so generously. We are also grateful to David Bradbury and Kamlyn Ramkissoon for their assistance.

# Additional information

### Funding

| Funder | Grant reference number | Author |
| --- | --- | --- |
| Wellcome | 101759/Z/13/Z | Eleanor A Maguire |
| Wellcome | 203147/Z/16/Z | Eleanor A Maguire |

The funders had no role in study design, data collection and interpretation, or the decision to submit the work for publication.

### Author contributions

Goffredina Spanò, Conceptualization, Formal analysis, Investigation, Methodology, Writing - original draft, Data scoring; Gloria Pizzamiglio, Investigation, Writing - review and editing, Data scoring; Cornelia McCormick, Thomas D Miller, Clive R Rosenthal, Writing - review and editing, Patient characterization; Ian A Clark, Writing - review and editing, Advice on data scoring; Sara De Felice, Writing - review and editing, Data scoring; Jamie O Edgin, Writing - review and editing, Advice on conceptualization; Eleanor A Maguire, Conceptualization, Formal analysis, Supervision, Funding acquisition, Methodology, Writing - original draft

### Author ORCIDs

Gloria Pizzamiglio (iD) https://orcid.org/0000-0002-3567-1344
Clive R Rosenthal (iD) https://orcid.org/0000-0002-5960-4648
Eleanor A Maguire (iD) https://orcid.org/0000-0002-9470-6324

### Ethics

Human subjects: Each participant gave written informed consent for participation in the study, for data analysis and for publication of the study results. 'Materials and methods' were approved by the University College London Research Ethics Committee (reference number: 6743/004).

### Decision letter and Author response

Decision letter https://doi.org/10.7554/eLife.56211.sa1
Author response https://doi.org/10.7554/eLife.56211.sa2

## Additional files

### Supplementary files

• Supplementary file 1. Summary of neuropsychological information. This table provides details of the neuropsychological profile (summary data and statistical analyses) of the patients across a range of cognitive tests, and indicates the selective nature of their memory loss.

• Supplementary file 2. Sleep quality of the patients and control participants. This table provides details (summary data and statistical analyses) of subjective measures of the general quality and pattern of sleep, as well as on objective measures of sleep-related breathing disorders, and sleep-wake patterns across one week. There were no significant differences between the patient and control participants on any measure.

• Transparent reporting form

### Data availability

The dream data for every participant for every measure are provided in the file Table 2-Source data 1.

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
