## [Decision Letter]

**Acceptance summary:**

This is a timely and elegant study exploring dreaming in patients with bilateral hippocampal damage due to LGI1-antibody-complex LE. Studies like this have a strong potential to inform us about the neural basis of dreaming, which remains largely unknown.

**Decision letter after peer review:**

Thank you for submitting your article "Dreaming with hippocampal damage" for consideration by *eLife*. Your article has been reviewed by three peer reviewers, one of whom is a member of our Board of Reviewing Editors, and the evaluation has been overseen by Laura Colgin as the Senior Editor. The following individual involved in review of your submission has agreed to reveal their identity: Jess Payne (Reviewer #3).

The reviewers have discussed the reviews with one another and the Reviewing Editor has drafted this decision to help you prepare a revised submission. In recognition of the fact that revisions may take longer than the two months we typically allow, until the research enterprise restarts in full, we will give authors as much time as they need to submit revised manuscripts.

Summary:

This is a timely and interesting study from the Maguire lab exploring the role of the hippocampus in dreaming. Four patients with bilateral hippocampal damage, due to LGI1-antibody-complex LE were tested at home using a tightly controlled forced awakening protocol informed by PSG. These patients are extremely rare, presenting with selective memory disturbances due to circumscribed hippocampal insult, which adds to the novelty of the study. The authors used an elegant experimental design including a habituation protocol and the use of PSG to guide awakenings in participants. Interestingly, the current study findings are in line with older papers by Torda from the 1960s emphasising dream abnormalities in amnesia, as opposed to more recent investigations which characterise dreaming as largely intact in such patients (Solms, Stickgold). While the reviewers were all enthusiastic about this study, a number of methodological concerns were nevertheless raised.

Essential revisions:

1) It was suggested that the novelty of the study was somewhat oversold given a number of studies that have explored dreaming in patients with extensive hippocampal damage. It is argued that these studies are flawed for several reasons, including because damage actually extended beyond the hippocampus and patients had broader cognitive impairments. This may be the case, but the clear presence of dreaming in these patients (e.g. Torda, 1969 studies; Stickgold et al., 2000) still shows that dreaming can persist in the face of bilateral hippocampal lesions. The current study is therefore not the first to test whether dreaming requires the hippocampus and such statements should be tempered accordingly.

2) Information on the timing of dream awakenings should be expanded in the Materials and methods. It is stated that participants were awoken after a solid 3min period of REM or NREM sleep, but how, specifically, was the timing and number of awakenings determined? How often were they awoken, and at what time of night? Were awakenings stopped when some maximum number was reached? Were REM and NREM reports collected at equivalent times of night? Was the clock time of awakenings equivalent between patients and controls? Further details would be helpful to address these comments.

3) One methodological weakness was the verbal interaction between the researchers and patients during dream reporting. Subtle differences in how questions are posed have a large effect on the quantity and quality of dream reports. Thus, it is ideal to use pre-recorded verbal prompts for all participants, avoiding systematic bias in the way that dream reports are elicited from patients vs from controls. In this regard, it could be a problem that participants "were occasionally prompted (e.g. Can you tell me more about that?) to obtain further information". While the number of prompts did not differ significantly between control and patient groups, the study is underpowered to detect all but the largest differences here. Numerically, control participants were prompted for more information more often than the patients. There also could have been subtle differences in the quality of interaction between researchers and participants that affected reporting.

4) It is reasoned that "If patients reported fewer dreams due to an inability to recollect what had just been in their mind, the two groups should differ in terms of the proportion of blank dreams". There exists no validated measure of dreaming other than participants' subjective report, and there is no evidence that participants can meaningfully distinguish between having had a dream that they cannot recall and having not had a dream at all. It is therefore of questionable validity to presume that "blank dreams" indicate that dreaming occurred and was forgotten, whereas "no dream" reports indicate lack of dreaming.

5) The very small sample of patients leads to a high risk of both Type II and Type I error. While a large sample is not feasible with this rare and difficult-to-test population, strong conclusions based on non-significant differences between groups should be avoided. For example, it is unclear whether patients are not just describing their dream experiences using fewer words than controls. Importantly, some of the methods used to score richness of detail (for example that used in Hassabis et al., 2007) are based on the number of words used – So differences in so-called "episodic richness" could be in part attributed to any differences in the mere ability of patients to describe their experience, rather than the production of dreaming itself.

6) A related point is whether it would be more informative to present the data as a case series rather than aggregating data from 4 extremely rare (and in some cases only 3) patients as a group. The figures currently present the group means, but individual data points should be included to enable the reader to see the spread of scores across patient and Control groups. A case series would enable the authors to further compare and contrast the case who had no dream reports and to potentially understand the heterogeneity across these patients. It may be that aggregating the dreaming performance across these patients masks important individual differences, as for some measures Controls appear to outperform patients by almost double (e.g., informative content) yet this is not statistically significant at the group level.

7) The authors argue that "if dreaming is unaffected by HC lesions, this would suggest that complex, associative, imagery-rich, spatio-temporal mental events can occur without hippocampal input, requiring a fundamental re-consideration of prominent theories of hippocampal function". It was suggested that the authors should be a bit more cautious in their language given that most HC output is believed to be blocked during REM sleep (which is where most imagery rich dreaming occurs). In fact, given that REM sleep dreaming may be mostly cortical (temporarily devoid of HC input), these results become even more interesting, because the REM sleep dreams obtained here were still degraded compared to controls. These issues may lie beyond the scope of the present study but the authors might find these points relevant for future work.

8) While the proportion of awakenings between NREM and REM were not significantly different between patients and controls, the reviewers were concerned regarding the number of dream reports solicited from REM sleep in general, and especially in the patients. Proportion of awakenings in REM in patients looks to be only about 20%. This is concerning given that REM sleep is where most genuine "dreams" occur (according to standard definitions of dreams as opposed to the mental content that can be obtained from NREM sleep). The authors should comment on this.

---

## [Author Response]

Essential revisions:1) It was suggested that the novelty of the study was somewhat oversold given a number of studies that have explored dreaming in patients with extensive hippocampal damage. It is argued that these studies are flawed for several reasons, including because damage actually extended beyond the hippocampus and patients had broader cognitive impairments. This may be the case, but the clear presence of dreaming in these patients (e.g. Torda, 1969 studies; Stickgold et al., 2000) still shows that dreaming can persist in the face of bilateral hippocampal lesions. The current study is therefore not the first to test whether dreaming requires the hippocampus and such statements should be tempered accordingly.

Apologies, we did not intend to imply that our study was the first to examine dreaming in hippocampal-damaged patients. Rather, we sought to note possible reasons for the disparity among extant findings, and to outline how our study helps to mitigate issues that may have influenced previous results. We have now gone through the entire manuscript and revised it to ensure that our intent is clearer and that the language is appropriately tempered. Examples of how we have done this include:

Abstract: “The hippocampus is linked with both sleep and memory, but there is debate about whether a salient aspect of sleep – dreaming – requires its input.”

Introduction: “While sleep dreaming has been examined previously in patients with hippocampal lesions, results are mixed, with some studies reporting repetitious and stereotyped dreams (Torda, 1969a,b; Stickgold et al., 2000), whereas others claim that hippocampal damage has no effect on dreaming (Solms, 1997; 2013). Possible reasons for these disparate findings include differences in the sleep stages sampled, the inclusion in several studies of patients with psychiatric conditions and below-average IQ and, in other instances, patients were interviewed at a point temporally remote from any dreaming that might have occurred, presenting a challenge for these patients who usually suffer from amnesia.”

“Here, we sought to mitigate the issues affecting previous studies while investigating the frequency, features and content of dreaming in rare patients with selective bilateral hippocampal damage and matched control participants.”

Discussion: “By studying rare patients with selective bilateral hippocampal damage we found that dream frequency was reduced compared to control participants, and the few dreams they had were much less episodic-like in nature and lacked content. This accords with previous studies that reported stereotyped dreaming in patients with brain damage that extended beyond the hippocampi (Torda, 1969a,b; Stickgold et al., 2000)…”

2) Information on the timing of dream awakenings should be expanded in the Materials and methods. It is stated that participants were awoken after a solid 3min period of REM or NREM sleep, but how, specifically, was the timing and number of awakenings determined? How often were they awoken, and at what time of night? Were awakenings stopped when some maximum number was reached? Were REM and NREM reports collected at equivalent times of night? Was the clock time of awakenings equivalent between patients and controls? Further details would be helpful to address these comments.

For all participants, awakenings occurred after sleep onset, throughout the night. We aimed to assess dreams from both NREM and REM sleep, and we therefore staggered awakenings at intervals that allowed for entry into these stages, or approximately between 30-90 minute intervals. For example, a participant may have been woken at 30 minutes, returned to sleep and was woken again at 90 minutes. A maximum of 10 awakenings were scheduled per night. Awakenings were not collected at pre-ordained points during the night (for example, after the third REM period for all participants), rather they were based on participants’ specific sleep architecture in order to maximize the number of reports collected. Every participant had a different sleep onset and duration, and so the awakenings were not scheduled at precisely the same clock time across participants. However, as shown in Supplementary file 2, the sleep quality of the patients and the controls was well-matched, and this included total sleep time, bedtime and midpoint of the night. We now include this additional information in the Materials and methods:

“For all participants, awakenings occurred after sleep onset, throughout the night. We aimed to assess dreams from both NREM and REM sleep, and we therefore staggered awakenings at intervals that allowed for entry into these stages, or approximately between 30-90 minute intervals. For example, a participant may have been woken at 30 minutes, returned to sleep and was woken again at 90 minutes. A maximum of 10 awakenings were scheduled per night. Awakenings were not collected at pre-ordained points during the night (for example, after the third REM period for all participants), rather they were based on participants’ specific sleep architecture in order to maximize the number of reports collected. Every participant had a different sleep onset and duration, and so the awakenings were not scheduled at precisely the same clock time across participants. However, as shown in Supplementary file 2, the sleep quality of the patients and the controls was well-matched, and this included total sleep time, bedtime and midpoint of the night.”

3) One methodological weakness was the verbal interaction between the researchers and patients during dream reporting. Subtle differences in how questions are posed have a large effect on the quantity and quality of dream reports. Thus, it is ideal to use pre-recorded verbal prompts for all participants, avoiding systematic bias in the way that dream reports are elicited from patients vs from controls. In this regard, it could be a problem that participants "were occasionally prompted (e.g. Can you tell me more about that?) to obtain further information". While the number of prompts did not differ significantly between control and patient groups, the study is underpowered to detect all but the largest differences here. Numerically, control participants were prompted for more information more often than the patients. There also could have been subtle differences in the quality of interaction between researchers and participants that affected reporting.

This is an issue into which we put a lot of thought prior to data collection, mindful of the point highlighted by the reviewers. We agree that uncontrolled probing could affect findings. However, we were concerned that pre-recorded probing could also influence results. For instance, the latter could come across as somewhat impersonal, be of limited utility if participants suspect it is automated, and could provoke confusion in people who are being woken from sleep in the middle of the night. To achieve controlled probing that was nevertheless sensitive and useful, the elicitation of dream reports in our study followed a structured protocol. This involved first asking participants to freely describe what was in their minds immediately after awakening. Whenever a participant’s response was not clear or only covered parts of the dream, the experimenter asked general follow-up questions, which could echo information already provided (e.g. “…It was a conversation happening in the locker room”. Can you give us any more specifics about the conversations or anything else that you recall?). Crucially, this probing never involved leading the participant, as can be observed in the examples now provided in new Figure 2.

This approach is very similar to that of well-established tasks that assess autobiographical memory recall (Levine et al., 2002) and scene imagination ability (Hassabis et al., 2007) during wake, where probing in this manner is widely accepted. We, therefore, believe that this parsimonious approach to probing did not have a substantive effect on our findings. We note that the numerical difference in the mean number of probes per awakening was small between groups [HPC 3.82 (1.75); CTL 4.22 (1.13)], as was the effect size (0.23; p=0.671) with no hint that a large difference was masked. We now describe our probing procedure in more detail in the Materials and methods:

“Participants were occasionally probed (e.g. Can you tell me more about that?) to obtain further information. Probing followed a structured protocol. This involved first asking participants to freely describe what was in their minds immediately after awakening. Whenever a participant’s response was not clear or only covered parts of the dream, the experimenter asked general follow-up questions, which could echo information already provided (e.g. “…It was a conversation happening in the locker room”. Can you give us any more specifics about the conversations or anything else that you recall?). Crucially, this probing never involved leading the participant, as can be observed in the examples provided in Figure 2. This approach is very similar to that of well-established tasks that assess autobiographical memory recall (Levine et al., 2002) and scene imagination ability (Hassabis et al., 2007) during wake, where probing in this manner is widely accepted.”

4) It is reasoned that "If patients reported fewer dreams due to an inability to recollect what had just been in their mind, the two groups should differ in terms of the proportion of blank dreams". There exists no validated measure of dreaming other than participants' subjective report, and there is no evidence that participants can meaningfully distinguish between having had a dream that they cannot recall and having not had a dream at all. It is therefore of questionable validity to presume that "blank dreams" indicate that dreaming occurred and was forgotten, whereas "no dream" reports indicate lack of dreaming.

We agree with this point, and in the revised manuscript have now significantly tempered the interpretation of this finding. We still allude to this result and retain the related data in new Table 2 because the distinction between no dreams and blank dreams is common in the literature and will be of interest to some readers.

“Perhaps the patients reported fewer dreams than control participants simply because they forgot any dreams they may have had. Three different types of awakening were evident: when participants reported a dream, when they stated they did not dream at all (no dream), and when they dreamt but could not recall the content (this is known as a blank dream). Although there was a significant difference between patients and controls for dreams (see above) and for no dreams, they did not differ in terms of the proportion of blank dreams, which was low for both groups. This suggests that patients could distinguish between situations when they did not dream and those when they dreamt but could not remember. However, no validated objective measure of dreaming exists, and this should be borne in mind when interpreting participants' subjective reports. It is notable that in previous studies, these particular patients could retain information over several minutes, including reporting on their daydreaming (e.g. McCormick et al., 2016, 2018), which speaks against a rapid decay of sleep mentation as an explanation for their reduced dream frequency.”

5) The very small sample of patients leads to a high risk of both Type II and Type I error. While a large sample is not feasible with this rare and difficult-to-test population, strong conclusions based on non-significant differences between groups should be avoided. For example, it is unclear whether patients are not just describing their dream experiences using fewer words than controls. Importantly, some of the methods used to score richness of detail (for example that used in Hassabis et al., 2007) are based on the number of words used – So differences in so-called "episodic richness" could be in part attributed to any differences in the mere ability of patients to describe their experience, rather than the production of dreaming itself.

We concur with the need for caution in interpreting null results in small samples, and that is why we are happy to implement the reviewers’ excellent suggestions of tempering the overall tone, as well as including individual patient data to facilitate their consideration as a case series (see more on this latter issue in the response to point 6).

The number of informative words per dream report did not differ significantly between the patient and control groups, although there was variability across the board. Nevertheless, adjudicating between the possibility of patients having a generic problem with expressing themselves verbally versus merely having little to describe because the few dreams they had were so impoverished, is challenging. The same issue pertains for assessments during wake. A number of studies addressed this concern during tasks involving the imagination of scenes or future scenarios, counterfactual thinking, describing the present and pictures of scenes. Bilateral hippocampal damage does not affect narrative construction or verbal descriptive ability (e.g. Race et al., 2011, 2013; Mullally et al., 2012; Mullally and Maguire, 2014; Miller et al., 2020). Considering specifically the patients in the current study, they too had no difficulty performing verbal fluency tests or other tasks where total word count was measured (Supplementary file 1). Given these general and specific findings, it is unlikely that the patients’ performance was driven by an underlying expressive verbal problem. We now consider this point in more detail in the revised manuscript:

“We first examined the number of informative words (see Materials and methods; Stickgold et al., 2001b) used in the dream narratives and, while the patients used fewer such words, overall there was no significant difference between the groups. Nevertheless, adjudicating between the possibility of patients having a generic problem with expressing themselves verbally versus merely having little to describe because the few dreams they had were so impoverished, is challenging. The same issue pertains for assessments during wake. A number of studies addressed this concern during tasks involving the imagination of scenes or future scenarios, counterfactual thinking, describing the present and pictures of scenes. Bilateral hippocampal damage does not affect narrative construction or verbal descriptive ability (e.g. Race et al., 2011, 2013; Mullally et al., 2012; Mullally and Maguire, 2014; Miller et al., 2020). Considering specifically the patients in the current study, they too had no difficulty performing verbal fluency tests or other tasks where total word count was measured (Supplementary file 1). Given these general and specific findings, it is unlikely that the patients’ performance was driven by an underlying expressive verbal problem.”

6) A related point is whether it would be more informative to present the data as a case series rather than aggregating data from 4 extremely rare (and in some cases only 3) patients as a group. The Figures currently present the group means, but individual data points should be included to enable the reader to see the spread of scores across patient and Control groups. A case series would enable the authors to further compare and contrast the case who had no dream reports and to potentially understand the heterogeneity across these patients. It may be that aggregating the dreaming performance across these patients masks important individual differences, as for some measures Controls appear to outperform patients by almost double (e.g., informative content) yet this is not statistically significant at the group level.

We have now re-worked the manuscript to accommodate this excellent suggestion as follows:

– Figure 1 has been expanded to constitute an overall summary of the experimental set-up, the number of awakenings, and the main group differences in terms of dream frequency and content. This permits an at-a-glance appreciation of the substantive aspects of the study, and may be useful for readers, particularly those who do not work in the field.

– We now include a new Table 1 which contains individual patient data for a range of variables including hippocampal volume and percent hippocampal volume loss relative to control participants.

– New Table 2 provides all of the dream-related variables as before, but has been expanded to include the individual data for each patient for every measure. We have now also uploaded the related Table 2-Source Data 1 file which contains the individual data for patients and controls. Along with new Table 1, new Table 2 permits interested readers to examine the patients as a series of case studies. We retain the formal between-group comparisons for those readers who will be interested in these analyses, while tempering them appropriately in the text.

– New Figure 2 provides an example dream report from each of the four patients and one from an example healthy control participant to further facilitate the case study approach. We selected those that precluded personal identification of participants, given data protection legislation.

The reviewers suggested we include the data for each patient in case they masked factors that might have influenced the results. Consideration of individual differences in such a small sample of rare patients is not possible. Nevertheless, these data de-emphasise several possible explanations for the results. For example, the findings are unlikely to have been driven by the extent of hippocampal volume loss – HCP3, who had no dreams at all, did not have the greatest hippocampal volume loss (this aligns with the wider literature where evidence of one-to-one mapping between hippocampal volume and function is mixed even among healthy controls). Sampling during specific sleep stages did not seem to have systematically affected performance. For instance, HCP3, with no dreams, had 40% of his awakenings during REM sleep, which should have maximised his opportunity for imagery-rich dreaming. By contrast, HCP1, despite having a low proportion of awakenings during REM had the highest dream frequency among the patients.

Overall, we believe these revisions render the manuscript more informative, whilst retaining clarity. We now preface the Results section as follows:

“Table 2 shows the group summary data and the results of the between-group statistical analyses for each of the measures that are described below. While, for the sake of economy, we present the findings in terms of these group comparisons, given the small sample of these rare patients, caution should be exercised in interpreting the results. We, therefore, also include the individual patient data in Tables 1 and 2 permitting the patients to be considered instead as a series of case studies.”

7) The authors argue that "if dreaming is unaffected by HC lesions, this would suggest that complex, associative, imagery-rich, spatio-temporal mental events can occur without hippocampal input, requiring a fundamental re-consideration of prominent theories of hippocampal function". It was suggested that the authors should be a bit more cautious in their language given that most HC output is believed to be blocked during REM sleep (which is where most imagery rich dreaming occurs). In fact, given that REM sleep dreaming may be mostly cortical (temporarily devoid of HC input), these results become even more interesting, because the REM sleep dreams obtained here were still degraded compared to controls. These issues may lie beyond the scope of the present study but the authors might find these points relevant for future work.

Whilst appreciating the interesting comment, consideration of this issue is beyond the scope of the present work, and would be better pursued using a bespoke experimental design in a future study. The simple point we were trying to make in the cited text is, we hope, relatively uncontroversial. That is, if patients with significantly reduced ability to experience complex, imagery-rich, spatio-temporal mental events during wake are able to have such experiences during sleep, then this would need to be accommodated within extant hippocampal theories, with further implications for understanding the role of cortical regions and hippocampal-cortical interactions in such mental events. We have now re-worded the text to better reflect our point:

“It remains uncertain, therefore, whether hippocampal integrity is necessary for dreaming to occur. If, as we predicted, bilateral hippocampal damage degrades dreaming, this would reinforce the link between dreaming and hippocampal-dependent processes such as memory, potentially moving us closer to an understanding of why we dream. By contrast, if hippocampal-damaged patients who have a reduced ability to experience complex, imagery-rich, spatio-temporal mental events during wake are nevertheless able to have such experiences during sleep, this would need to be explicitly accounted for within theories of hippocampal function.”

8) While the proportion of awakenings between NREM and REM were not significantly different between patients and controls, the reviewers were concerned regarding the number of dream reports solicited from REM sleep in general, and especially in the patients. Proportion of awakenings in REM in patients looks to be only about 20%. This is concerning given that REM sleep is where most genuine "dreams" occur (according to standard definitions of dreams as opposed to the mental content that can be obtained from NREM sleep). The authors should comment on this.

Dreams (and sleep-related visualisations) have been recorded in all sleep stages, even stage 1. REM is the stage in which people report “typically” bizarre or emotional content, but dreams with spatial content are also reported from NREM awakenings. Hence, we felt it was necessary to assess NREM and REM sleep in order to capture dream content that might be affected by hippocampal damage. Also, as noted by the reviewers, there was no significant difference between the patients and control participants in terms of the number of awakenings from NREM and REM. Moreover, and as now detailed on new Table 2 including its legend, the proportion of awakenings during REM in patients may be an underestimate. For HCP1, during 20% of his awakenings towards the end of the night, the EEG cap stopped functioning and so designation to NREM or REM sleep was not possible. Hence, it could be that his zero score for REM awakenings is an underestimate, given that REM is more common in the latter part of the night. If we assume that these awakenings occurred during REM, the patient group mean would then be 28%, which is even more similar to the controls (33%).

As alluded to in the response to point 7 above, the current study was not designed to investigate differences between NREM and REM dreaming, and so our data cannot speak to ongoing debates in that realm. Our goal was to sample the night’s sleep, which typically comprises ~20% REM, in as similar a manner as possible across the patient and control groups.